# Statin-mediated disruption of Rho GTPase prenylation and activity inhibits respiratory syncytial virus infection

Manpreet Malhi[1,2], Michael J. Norris[3,4], Wenming Duan[4], Theo J. Moraes[3,4,5] & Jason T. Maynes [1,2,6✉]

Respiratory syncytial virus (RSV) is a leading cause of severe respiratory tract infections in children. To uncover new antiviral therapies, we developed a live cell-based high content screening approach for rapid identification of RSV inhibitors and characterized five drug classes which inhibit the virus. Among the molecular targets for each hit, there was a strong functional enrichment in lipid metabolic pathways. Modulation of lipid metabolites by statins, a key hit from our screen, decreases the production of infectious virus through a combination of cholesterol and isoprenoid-mediated effects. Notably, RSV infection globally upregulates host protein prenylation, including the prenylation of Rho GTPases. Treatment by statins or perillyl alcohol, a geranylgeranyltransferase inhibitor, reduces infection in vitro. Of the Rho GTPases assayed in our study, a loss in Rac1 activity strongly inhibits the virus through a decrease in F protein surface expression. Our findings provide new insight into the importance of host lipid metabolism to RSV infection and highlight geranylgeranyltransferases as an antiviral target for therapeutic development.

[1] Department of Biochemistry, University of Toronto, Toronto, ON, Canada. [2] Program in Molecular Medicine, The Hospital for Sick Children, Toronto, ON, Canada. [3] Department of Laboratory Medicine and Pathobiology, University of Toronto, Toronto, ON, Canada. [4] Program in Translational Medicine, The Hospital for Sick Children, Toronto, ON, Canada. [5] Department of Paediatrics, Division of Respiratory Medicine, The Hospital for Sick Children, Toronto, ON, Canada. [6] Department of Anesthesia and Pain Medicine, The Hospital for Sick Children, Toronto, ON, Canada. ✉email: jason.maynes@sickkids.ca

Respiratory syncytial virus (RSV) is an enveloped, negative-sense RNA virus of the family *Pneumoviridae*[1]. It is the leading cause of acute respiratory infections (ARIs) necessitating hospitalization in children, with an estimated disease burden comparable to that of seasonal influenza[2]. Globally, the virus contributes to 33.1 million annual cases of ARIs resulting in 3.2 million hospital admissions and 59,600 deaths in children under the age of 5[3]. The clinical manifestations of RSV range from mild infection of the upper respiratory tract, characterized by cough, rhinitis, and occasional fever, to severe infection of the lower respiratory tract resulting in bronchiolitis, pneumonia, or acute respiratory failure[4]. To date, no RSV vaccines have been licensed for patient use and current treatment options are limited to supportive care (e.g. administration of oxygen, intravenous fluids, or mechanical ventilation)[5]. Viral infection can be prevented by palivizumab immunoprophylaxis; however, this approach requires repeated monthly injections each viral season and is not feasible for general use in the population[6]. Given the substantial burden of RSV infection, development of treatments for the virus is imperative.

Traditional strategies for antiviral drug design have primarily focused on direct inhibition of virus-encoded factors. However, it is equally important to consider that viruses rely on the host cell environment for replication, protein expression, and assembly of progeny virion. This has led to the emergence of host-directed antivirals as an alternative therapeutic paradigm[7,8]. Metabolites of the mevalonate pathway, including cholesterol and isoprenoids, have been identified as host factors critical to RSV infection and are potential targets for drug intervention. Cholesterol is a key component of lipid rafts, which serve as a docking point in the plasma membrane for the assembly and release of viral particles[9–12]. Furthermore, the lipid raft content of enveloped viruses, including RSV, has been correlated to virion stability[13]. Isoprenoids are required for protein prenylation, a post-translational modification whereby farnesyl or geranylgeranyl lipids derived from mevalonate are covalently bound to conserved cysteine residues near the C-terminus of proteins[14]. During viral entry, assembly, and budding, the directional movement of RSV components is dependent on interactions with the host cytoskeleton[15–17]. Dynamics of the actin cytoskeleton are mediated by the Rho family of small GTPases (RhoA, Rac1, Cdc42), which rely on prenylation to regulate their subcellular localization and activity[18,19].

To address the healthcare burden associated with RSV infection, we developed and conducted a high content screen to identify inhibitors of the virus. Pathway analysis of the hits from our screen revealed a strong functional enrichment in lipid and steroid metabolic pathways. We explored the role of these host pathways in RSV infection through statin-mediated modulation. Statins are reversible inhibitors of 3-hydroxy-3-methylglutaryl-CoA reductase (HMGCR), the rate-limiting enzyme of the mevalonate pathway. Our results indicate that statins inhibit post-entry events in the replicative cycle of RSV through a combination of cholesterol and isoprenoid-mediated effects, the latter involving a disruption in virus-induced Rho GTPase activity due to a loss in protein prenylation.

## Results

### Development of a live cell-based high content screen to identify inhibitors of RSV infection

To identify inhibitors of RSV infection, we designed a fully automated image acquisition and analysis pipeline to screen compounds against HEp-2 cells infected with recombinant RSV engineered to express GFP (RSV-GFP) (Fig. 1a). The recombinant virus was created by insertion of the GFP coding sequence prior to the first viral gene in a full-length cDNA copy of the RSV A2 genome, with the construct generating near-parental viral titers[20,21]. Automated image acquisition of infected cells was performed using the Cellomics ArrayScan VTI HCS Reader (Fig. 1a). Infection rates were determined from automated and calibrated image segmentation by taking the ratio of nuclei within GFP-positive area (i.e. infected cells) to total nuclei (i.e. all cells) (Fig. 1b). To increase hit specificity, changes to mean nuclear fluorescent intensity were quantified as a measure of nuclear condensation and apoptosis due to potential drug toxicity[22]. We screened a total of 2400 compounds from the MicroSource SPECTRUM Collection, largely consisting of FDA-approved drugs and other known bioactive compounds. A mean Z-factor of $0.65 \pm 0.19$ was calculated from the tested plates, indicative of a reasonable statistical effect size (Fig. 1c).

From the screen, 60 compounds were identified as hits (2.5% hit rate) based on the criteria of inhibiting RSV infection >50% with low cellular toxicity (i.e. altering mean nuclear intensity <10%) (Fig. 1d and Supplementary Table 1). The majority of hits could be grouped into one of five general functional drug classes, including statins, steroid hormones, cardiac glycosides, anti-infectives, and flavonoids (Table 1). Selected hits from each class of drugs were tested over a range of concentrations to establish dose–response curves for validation purposes (Supplementary Fig. 1). Cardiac glycosides exhibited the most potent antiviral activity, inhibiting RSV with an effective concentration ($EC_{50}$) in the low nanomolar range (Table 1). Four unique statins (atorvastatin, fluvastatin, pitavastatin, and simvastatin) were also found to inhibit RSV at low micromolar concentrations (Table 1 and Supplementary Table 1).

### Functional enrichment of lipid and steroid metabolic pathways among screening hits

Using the target proteins of hits identified from our drug screen, functional enrichment analysis was carried out in PANTHER to uncover requisite host pathways for RSV[23]. A list of 133 unique proteins was compiled using compounds from our screen with known and characterized molecular targets (Supplementary Table 2). Top ranking GO biological processes were found to be related to general drug metabolism and cellular responses to xenobiotic stimuli (Supplementary Table 3), consistent with the addition of chemical compounds from a drug library. However, the pathways with the highest functional enrichment related to normal biological processes involved lipid and steroid metabolism (Supplementary Table 3). Concordantly, the ER and plasma membrane, both key sites of lipid dynamics, were among the most enriched GO cellular components (Supplementary Table 3). Visual interactome analysis was performed in ClueGO on the most significant biological processes and cellular components ($P < 1E{-}10$) (Fig. 1e, f). Steroid and isoprenoid metabolism emerged as major hubs within the interactome for enriched GO biological processes (Fig. 1e). Furthermore, the ER, plasma membrane, and caveolae were found to be hubs among enriched GO cellular components (Fig. 1f). Together, these results demonstrate the importance of lipid metabolism as a druggable host pathway utilized by RSV during its replicative cycle.

### Statins dose-dependently inhibit RSV infection of HEp-2 cells and well-differentiated PNECs

The informatic analysis of molecular targets corresponding to our drug screen hits highlighted the significance of lipid metabolic pathways, which are readily modulated by the statin class of drugs. To validate the effects of host lipid metabolism and statins on RSV, we further studied the antiviral activity of two representative, lipophilic statins ($\log P > 4.0$) identified as hits (atorvastatin and simvastatin). Hydrophilic statins ($\log P < 1.0$), including rosuvastatin and

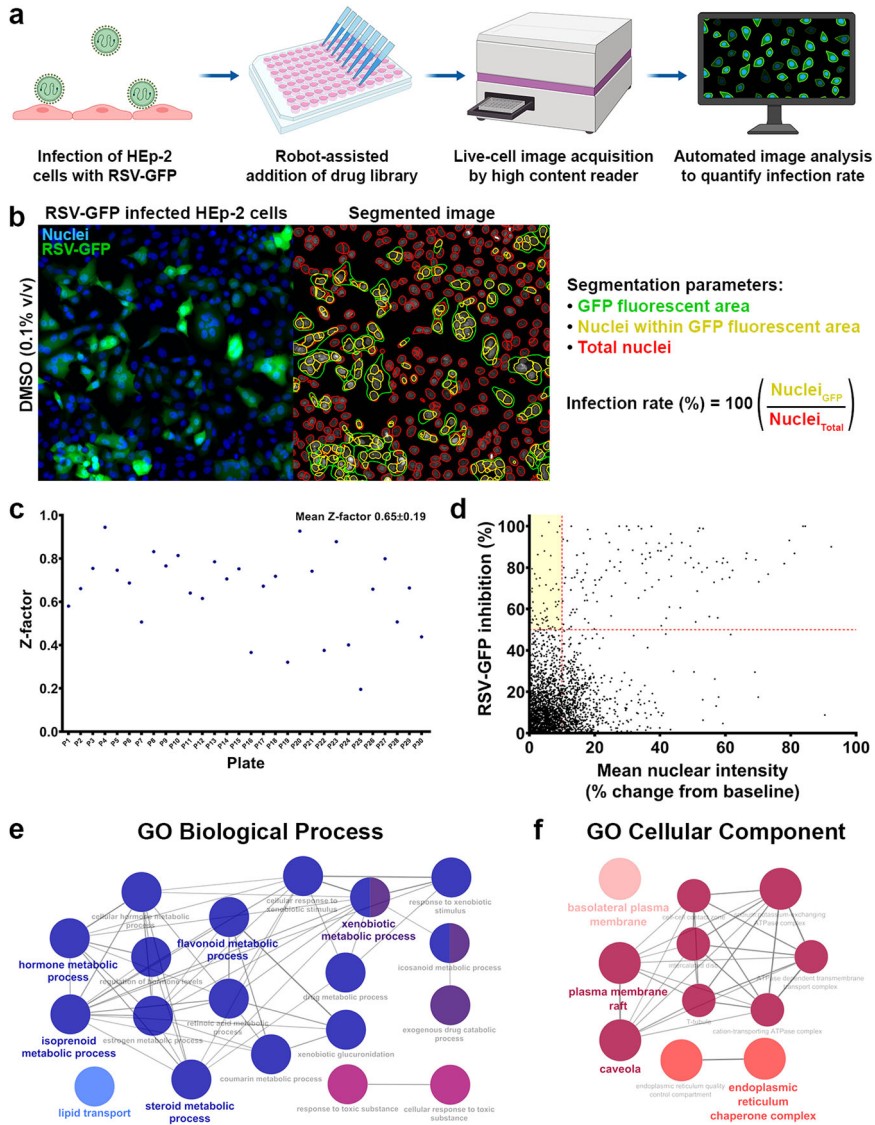

**Fig. 1 High-throughput quantification of RSV infection rates by automated imaging and analysis identifies RSV inhibitors and host pathways targeted by the virus. a** Methodology of screening for RSV inhibitors. HEp-2 cells are infected with recombinant RSV expressing green fluorescent protein (RSV-GFP) and treated 2 h post-viral exposure with test compounds. Live cell image acquisition is performed 48 h p.i. on the Cellomics ArrayScan VTI HCS Reader and automated image analysis is performed in CellProfiler. **b** Image segmentation is used to quantify infection rate from the ratio of nuclei within GFP-fluorescent area to total nuclei. This quantification approach accounts for in vitro syncytium formation induced by RSV, whereby a single infected cell may arise from the fusion of multiple cells. **c** Per-plate calculation of Z-factors to assess assay robustness. A mean Z-factor 0.65 ± 0.19 was determined for all tested plates. **d** Inclusion criteria of >50.0% RSV inhibition and <10.0% change in mean nuclear intensity were used to identify 60 hits from 2400 screened compounds (as indicated by the highlighted quadrant). **e**, **f** ClueGO network analysis of biological processes and cellular components enriched among the molecular targets of RSV inhibitors identified from drug screening. Steroid, isoprenoid, and other lipid metabolic processes are highly represented among screening hits. Additionally, the endoplasmic reticulum, plasma membrane, and caveolae were found to be notable sites of action in the cell during infection. Only pathways with a significance of $P < 1E−10$ are shown.

pravastatin, were not observed to inhibit RSV in our screen. HEp-2 cells are highly permissive to RSV and a common cell model for infection. Treatment of RSV-GFP infected HEp-2 cells with atorvastatin (3.5 µM) or simvastatin (1.5 µM) 2 h post-viral exposure caused a visible reduction in GFP-fluorescent area and the number of RSV-induced syncytia relative to DMSO control-treated cells (0.1% v/v) (Fig. 2a). Dose-dependent inhibition of RSV-GFP was observed for both atorvastatin and simvastatin treatment, with an $EC_{50}$ for each drug at 2.6 and 1.0 µM, respectively (Fig. 2b, c). Measurement of cellular viability by MTT assay showed no significant difference from baseline at $EC_{50}$ for either drug, although a significant reduction in viability occurred at doses greater than 10.0 µM (Fig. 2b, c).

Statin inhibition of RSV was verified in an additional model of the human airway epithelium, utilizing primary human nasal epithelial cells (PNECs) grown on an air–liquid interface (Fig. 2d). When cultured in this manner, these cells differentiate into a pseudostratified multilayer tissue highly representative of human pulmonary epithelium[24]. Furthermore, the culture conditions simulate the polarized environment of airway epithelium in vivo. Dose-dependent reduction in GFP signal was observed 72 h post-infection (h p.i.) over a range of atorvastatin concentrations that were consistent with our findings in HEp-2 cells (Fig. 2b, e). RSV-GFP infection was reduced by 67.4 ± 11.5% at the highest tested dose of atorvastatin (5.0 µM) ($P < 0.05$) (Fig. 2e). Cellular viability remained intact following drug addition, as determined by overall

**Table 1 Summary of drug classes found to inhibit RSV infection.**

| Drug class | Drug name | EC$_{50}$ (µM) | HillSlope | $R^2$ |
|---|---|---|---|---|
| Statins | Atorvastatin | 2.558 | 2.112 | 0.9556 |
| | Simvastatin | 1.044 | 2.704 | 0.9195 |
| Steroid hormones | 17β-Estradiol | 1.622 | 3.809 | 0.9209 |
| | 19-Norethindrone | 0.705 | 1.817 | 0.9591 |
| Cardiac glycosides | Digoxin | 0.026 | 1.404 | 0.9876 |
| | Digitoxin | 0.011 | 1.229 | 0.9415 |
| Anti-infectives | Niclosamide | 0.443 | 8.875 | 0.9509 |
| | Valinomycin | 0.039 | 10.96 | 0.9443 |
| Flavonoids | 6-Hydroxyflavone | 0.705 | 1.874 | 0.9600 |

HEp-2 cells were infected with RSV-GFP (MOI 0.5) and treated 2 h post-viral exposure with a range of concentrations for each drug to generate dose–response curves. See Supplementary Fig. 1 for individual dose–response curves and Supplementary Table 1 for a complete list of all 60 drug screen hits.

microscopic appearance of the cells and maintenance of barrier integrity (i.e. the apical culture surface remained dry).

**Statins inhibit the production of infectious virus.** To gain a better understanding of the viral processes targeted by statins, we conducted a time-of-drug addition assay[25]. Prior studies in HEp-2 cells have shown that RSV mRNA and proteins can be detected 4–6 h p.i., while assembly is initiated roughly 10–12 h p.i., and mature viral filaments can be observed at the cell surface 16–18 h p.i.[26–28]. Atorvastatin (3.5 µM) was added to RSV-GFP-infected HEp-2 cells (MOI 2.0) at discrete timepoints pre- or post-infection, and infection was quantified 24 h p.i to approximately capture a single cycle of infection. No difference in antiviral effect was observed when atorvastatin was added 2 h pre-infection vs. 2–6 h post-infection, indicating cell entry was unaffected by the drug (Fig. 3a). A loss in antiviral effect occurred when atorvastatin was added 8–22 h p.i., consistent with inhibition of post-entry events in the viral life cycle (e.g. viral replication, assembly, or release) (Fig. 3a). To verify this finding, a traditional plaque assay was performed to quantify infectious virus production from the supernatants of statin-treated RSV A2-infected HEp-2 cells. Relative to cells treated with DMSO control (0.1% v/v), atorvastatin (3.5 µM) and simvastatin (1.5 µM) caused an 8.3- and 9.1-fold decrease in the production of infectious progeny virions, respectively ($P < 0.01$ for both statins) (Fig. 3b). Lysates were also collected from RSV A2-infected HEp-2 cells (MOI 0.5) 24 h p.i. to probe viral protein expression by western blot. Atorvastatin (3.5 µM) and simvastatin (1.5 µM) treatment 2 h post-viral exposure caused a decrease in total viral protein levels 24 h p.i. (Fig. 3c and Supplementary Fig. 2). Changes to the expression and localization of the viral fusion glycoprotein (F) and nucleoprotein (N) were visualized by high-resolution confocal microscopy. Atorvastatin treatment perturbed surface expression of the F protein, causing it to instead localize perinuclearly (i.e. to the ER–Golgi network) (Fig. 3d and Supplementary Fig. 3). The N protein localized to cytoplasmic viral inclusions (Fig. 3d). Surface expression of viral glycoproteins has previously been shown to be an essential step in viral assembly[29,30], and is consistent with our observation that statins inhibit post-entry events in the replicative cycle of RSV.

**The antiviral effects of statins are mediated by the mevalonate pathway.** The primary cellular action of statins is to block the conversion of HMG-CoA to mevalonate by inhibiting HMGCR. It was necessary to confirm whether the antiviral effects of statins

were due to inhibition of its known molecular target, as opposed to direct interaction of the drug with virus-encoded factors or other off-target effects. To address this, RSV A2-infected HEp-2 cells (MOI 0.5) were co-treated 2 h post-viral exposure with statins and mevalonolactone (a membrane-permeable ester of mevalonate). Exogenous addition of mevalonolactone restores mevalonate levels and pathway activity downstream of HMGCR in the presence of statin[31]. Treatment with atorvastatin (3.5 µM) or simvastatin (1.5 µM) alone reduced viral protein intensity across the total cell population (as quantified by F and N immunofluorescence), indicative of viral inhibition ($P < 0.01$ for both statins) (Fig. 4a–c). Co-treatment with mevalonolactone (1 mM) was found to restore viral protein intensity to DMSO (0.1% v/v) baseline in statin-treated cells ($P > 0.05$ for both statins) (Fig. 4a–c). This finding demonstrates the requirement of mevalonate biosynthetic activity for RSV infection, and confirms the antiviral effects of statins are mediated through this pathway.

**Cholesterol depletion disrupts surface localization of the F protein.** Cholesterol and isoprenoids are key downstream metabolites in the mevalonate pathway whose synthesis is inhibited by statins. In order to examine cholesterol-specific effects on RSV, HEp-2 cells were depleted of cholesterol using methyl-β-cyclodextrin (MBCD). Total cholesterol was quantified by filipin III fluorescence. Relative to DMSO control (0.1% v/v), atorvastatin (3.5 µM) decreased total cellular cholesterol by 16.2 ± 3.6% and simvastatin (1.5 µM) decreased total cholesterol by 15.8 ± 2.9% after 48 h of treatment ($P < 0.05$ for both statins) (Fig. 4d), consistent with previous findings in other cell models[32,33]. Chronic exposure to 1.0 mM MBCD for 48 h reduced total cholesterol by 20.6 ± 5.0% ($P > 0.05$ compared to statin treatment; $P < 0.01$ compared to DMSO treatment), whereas the addition of a 30-min acute pre-treatment step with 10.0 mM MBCD reduced total cholesterol by 76.2 ± 5.1% ($P < 0.01$ compared to DMSO and statin treatment) (Fig. 4d). Exposure to the lower dose of MBCD (1.0 mM) induced approximately the same degree of cholesterol depletion as statins in HEp-2 cells. We sought to determine the difference in antiviral effect between low-dose MBCD (moderate cholesterol depletion) and high-dose MBCD (high cholesterol depletion). HEp-2 cells were infected with RSV A2 (MOI 0.5) and the F protein was quantified by immunofluorescence 48 h p.i. Moderate cholesterol depletion with 1.0 mM MBCD was found to reduce F intensity by 21.6 ± 3.1% ($P < 0.01$) and high cholesterol depletion with 10.0 mM MBCD reduced F intensity by 31.1 ± 2.1% ($P < 0.01$), relative to DMSO control (0.1% v/v) (Fig. 4e). By comparison, atorvastatin (3.5 µM) and simvastatin (1.5 µM) reduced F intensity by 57.5 ± 4.8% and 66.0 ± 1.0%, respectively ($P < 0.01$ for both statins) (Fig. 4e). Changes to viral proteins in the cell were visualized by high-resolution confocal microscopy. Cholesterol depletion perturbed surface localization of the F protein, causing it to instead localize perinuclearly (Fig. 4f). The N protein was localized to cytoplasmic viral inclusions (Fig. 4f). Although cholesterol depletion and statin treatment both inhibited surface localization of the F protein, statin treatment had a relatively greater inhibitory effect on the virus. This finding was suggestive of a role for non-sterol mevalonate pathway metabolites (e.g. isoprenoids) in the antiviral mechanism of statins.

**Protein geranylgeranylation is specifically required for RSV infection.** Isoprenoids are products of the mevalonate pathway required for post-translational prenylation of proteins. We next wanted to determine whether a loss in protein prenylation contributed to statin antiviral activity. RSV A2-infected HEp-2 cells

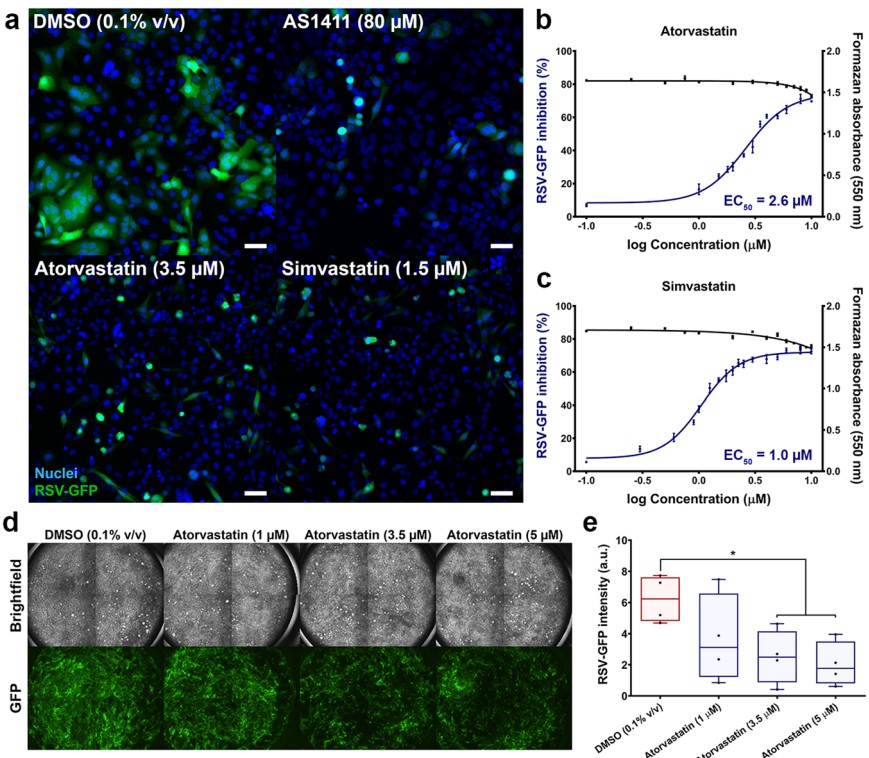

**Fig. 2 Statins dose-dependently inhibit RSV infection of HEp-2 cells and well-differentiated PNECs.** Live fluorescent imaging of cells infected with RSV-GFP demonstrates viral inhibition following statin treatment, as indicated by decreased GFP fluorescence and reduced syncytium formation. **a** Representative images of infected HEp-2 cells acquired 48 h p.i. Statins were added 2 h post-viral exposure. DMSO was used as a vehicle control and AS1411 was included as a positive control for inhibiting infection. Scale bars = 100 μm. **b, c** Dose-dependent inhibition of RSV-GFP (MOI 0.5) by statins was observed in HEp-2 cells with effective concentrations ($EC_{50}$) of 2.6 and 1.0 μM for atorvastatin and simvastatin, respectively (curves shown in blue). No significant changes to cellular viability were observed from baseline at $EC_{50}$ for each drug, as quantified by MTT assay (curves shown in black). Values are reported as means ± SEM ($n = 3$ independent measurements). **d** Human PNECs cultured on an air–liquid interface develop a ciliated phenotype which models the in vivo respiratory epithelium. Shown are representative images of well-differentiated PNECs acquired 72 h after apical surface infection with RSV-GFP (MOI 0.01). Atorvastatin was added to the basal medium at each specified concentration 2 h post-viral exposure. **e** Atorvastatin dose-dependently reduces RSV-GFP infection of WD-PNECs, with significant reductions observed at 3.5 and 5 μM ($n = 4$ PNEC donors). Box plots indicate the complete range (whiskers), interquartile range (box), and median (line) of measurements (a.u. arbitrary units). Statistics were calculated by a one-way ANOVA with a Dunnett's multiple comparisons post hoc test (*$P < 0.05$).

(MOI 0.5) were co-treated 2 h post-viral exposure with statins and isoprenoids, farnesyl pyrophosphate (FPP) or geranylgeranyl pyrophosphate (GGPP). Exogenous addition of GGPP (10.0 μM), but not FPP (10.0 μM), restored F protein intensity in the presence of atorvastatin (3.5 μM) and simvastatin (1.5 μM) by 33.4 ± 3.5% and 21.5 ± 4.0%, respectively ($P < 0.01$ compared to statin only) (Fig. 5a, b). Similarly, N protein intensity was restored by GGPP (10.0 μM), but not FPP (10.0 μM), in the presence of atorvastatin (3.5 μM) and simvastatin (1.5 μM) by 50.4 ± 1.6% and 44.0 ± 3.5%, respectively ($P < 0.01$ compared to statin only) (Fig. 5a, c).

Prenyltransferases are the enzymes responsible for covalent addition of isoprenoids to proteins[34]. Perillyl alcohol (PA), an inhibitor of geranylgeranyltransferase, dose-dependently reduced RSV A2 infection of HEp-2 cells whereas FTI-276, an inhibitor of farnesyltransferase, did not cause any significant changes to infection (Fig. 5d, e). At the maximum tested dose, PA (5.0 mM) reduced F protein intensity by 84.3 ± 1.6% ($P < 0.01$) and N protein intensity by 82.7 ± 3.3% ($P < 0.01$), relative to DMSO control (0.1% v/v) (Fig. 5d, e). Both inhibitors were tested at subtoxic concentrations exceeding their respective $IC_{50}$ to ensure maximal drug effect was detected[35,36]. These results highlight the importance of protein geranylgeranylation to RSV infection, and demonstrate how this post-translational modification can be targeted by small molecules to inhibit the virus.

**Statins mitigate RSV-induced increases to Rho GTPase prenylation and activity.** To measure changes to the prenylome by RSV and statins, we employed a click chemistry-based approach to isolate prenylated proteins (Fig. 6a). HEp-2 cells were cultured in the presence of azide-modified isoprenoids, farnesyl alcohol azide (FAA) or geranylgeranyl alcohol azide (GGAA)[37,38]. Following RSV A2 infection (MOI 0.5), lysates were collected 48 h p.i. and prenylated proteins were conjugated to biotin-PEG4-alkyne by a copper-catalyzed click chemistry reaction. All prenylated proteins were then visualized by western blot using a streptavidin-HRP probe. RSV infection caused an increase in total protein prenylation (both farnesylation and geranylgeranylation) relative to uninfected cells (Fig. 6b). Notably, we observed a dramatic virus-induced increase to the prenylation of ~20.0 kDa proteins (Fig. 6b) known to correspond to small prenylated GTPases based on bioinformatic and practical characterizations of the eukaryotic prenylome[39,40]. Treatment with atorvastatin (3.5 μM) was found to largely mitigate these virus-induced increases to small GTPase prenylation (Fig. 6b). The Rho family of GTPases are well-described host factors for viral infection, and the prenylation of these proteins is important for regulating their subcellular localization and activity[18,19,41]. Therefore, we measured active, GTP-bound Rho GTPase levels in RSV A2-infected HEp-2 lysates (MOI 0.5) with or without statin treatment. Accordingly, we found a decrease in GTP-bound

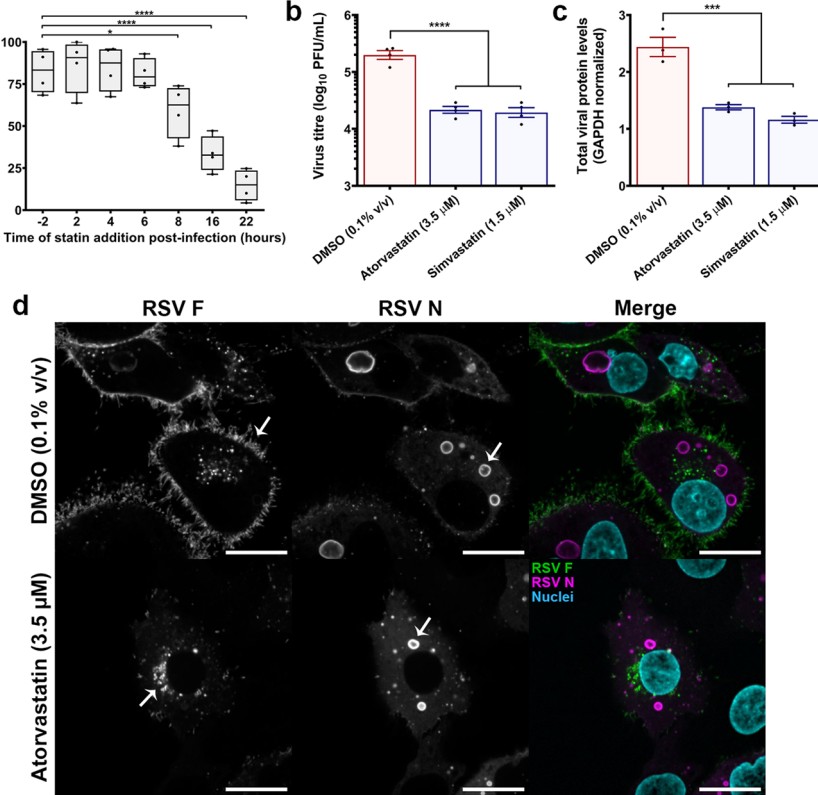

**Fig. 3 Statins inhibit the production of infectious virus. a** The actions of statins on the replicative cycle of RSV were probed by a time-of-drug addition assay. Atorvastatin (3.5 μM) was added to RSV-GFP-infected HEp-2 cells (MOI 2.0) at discrete timepoints pre- or post-infection, and infection rate was quantified 24 h p.i. No difference in antiviral effect was observed between 2-h pre-treatment and 2- to 6-h post-treatment, indicating cell entry was unaffected by statins. A loss in antiviral effect occurred when statins were added 8–22 h p.i., consistent with inhibition of post-entry events. Box plots indicate the complete range (whiskers), interquartile range (box), and median (line) of measurements. The y-axis denotes atorvastatin inhibition of RSV-GFP normalized to its maximum effect (i.e., % of maximum statin inhibition) (n = 4 independent measurements). **b** Infectious virus production was quantified by plaque assay from the supernatants of RSV A2-infected HEp-2 cells +/− atorvastatin (3.5 μM) or simvastatin (1.5 μM). Statin treatment caused a significant decrease in the production of infectious progeny virions. Values are reported as means ± SEM (n = 4 independent measurements). **c** Western blot analysis of lysates collected from RSV A2-infected HEp-2 cells (MOI 0.5) +/− atorvastatin (3.5 μM) or simvastatin (1.5 μM) 24 h p.i. Statin treatment caused a decrease in total viral protein levels. Blots were stained using a pan-RSV polyclonal antibody (B65860G) that detects all viral antigens (see Supplementary Fig. 2). GAPDH controls were processed on a parallel blot with the same samples and protein load. Values are reported as means ± SEM (n = 3 cell lysates). **d** High-resolution confocal microscopy of the F (green) and N (magenta) proteins in RSV A2-infected HEp-2 cells (MOI 0.5). Atorvastatin treatment decreases F surface expression, and N is localized to cytoplasmic inclusions (as indicated by the arrows). Scale bars = 20 μm. Statistics were calculated by a one-way ANOVA with a Dunnett's multiple comparisons post hoc test (*$P < 0.05$, ***$P < 0.001$, ****$P < 0.0001$).

RhoA and Rac1 levels in atorvastatin-treated infected samples, relative to DMSO control (0.1% v/v) (Fig. 6c). Taken together, our findings demonstrate that statins block virus-induced increases to the prenylation and activity of Rho GTPases.

**Rac1 inhibition disrupts surface expression of the F protein and infectious virus production**. To determine whether Rho GTPase inhibition decreased the production of infectious virus, as was shown for statins, a plaque assay was conducted on supernatants collected from RSV A2-infected HEp-2 cells treated with small-molecule inhibitors of RhoA (Rhosin), Rac1 (EHT 1864), and Cdc42 (ML 141) 2 h post-viral exposure. Each inhibitor has previously been shown to be selective for its target Rho GTPase[42–44]. Relative to DMSO control (0.1% v/v), treatment with the Rac1 inhibitor EHT 1864 (20.0 μM) caused a 10.1-fold reduction in infectious virus production ($P < 0.01$) (Fig. 7a). We further verified this finding by our in vitro immunofluorescence assay in RSV A2-infected HEp-2 cells (MOI 0.5). Rac1 inhibition by EHT 1864 (20.0 μM) reduced F and N protein intensity by 89.8 ± 0.8% and 70.4 ± 1.6%, respectively, across the total cell population 48 h p.i. ($P < 0.01$) (Fig. 7b, c). Treatment with a Rho

GTPase activator (CN04, 1.0 μM) elicited the opposite effect, instead enhancing RSV infection, promoting syncytial formation, and increasing F and N intensity by 56.4 ± 12.7% and 67.5 ± 4.7%, respectively ($P < 0.01$) (Fig. 7b, c). Co-treatment of atorvastatin (3.5 μM) with CN04 (1.0 μM) negated the pro-viral effects of Rho GTPase activation (Fig. 7b, c). Inhibition of Rac1 by EHT 1864 (20.0 μM) was also found to decrease total viral protein levels by western blot 24 h p.i. (Supplementary Fig. 2). Confocal microscopy of EHT 1864-treated RSV A2-infected HEp-2 cells (MOI 0.5) revealed that Rac1 inhibition resulted in a loss of F glycoprotein surface expression, similar to statin-treated cells (Figs. 3d and 7d). N was expressed and localized to cytoplasmic viral inclusions (Fig. 7d). Our results demonstrate that Rho GTPase activity facilitates viral infection, and inhibition of the Rho GTPase, Rac1, is detrimental to the replicative cycle of RSV.

**Discussion**
Discovery of novel antivirals has previously been limited by the low throughput of traditional methods for quantifying viral infection (e.g. plaque assays). More recently, the use of recombinant viruses with fluorescent or bioluminescent reporters has

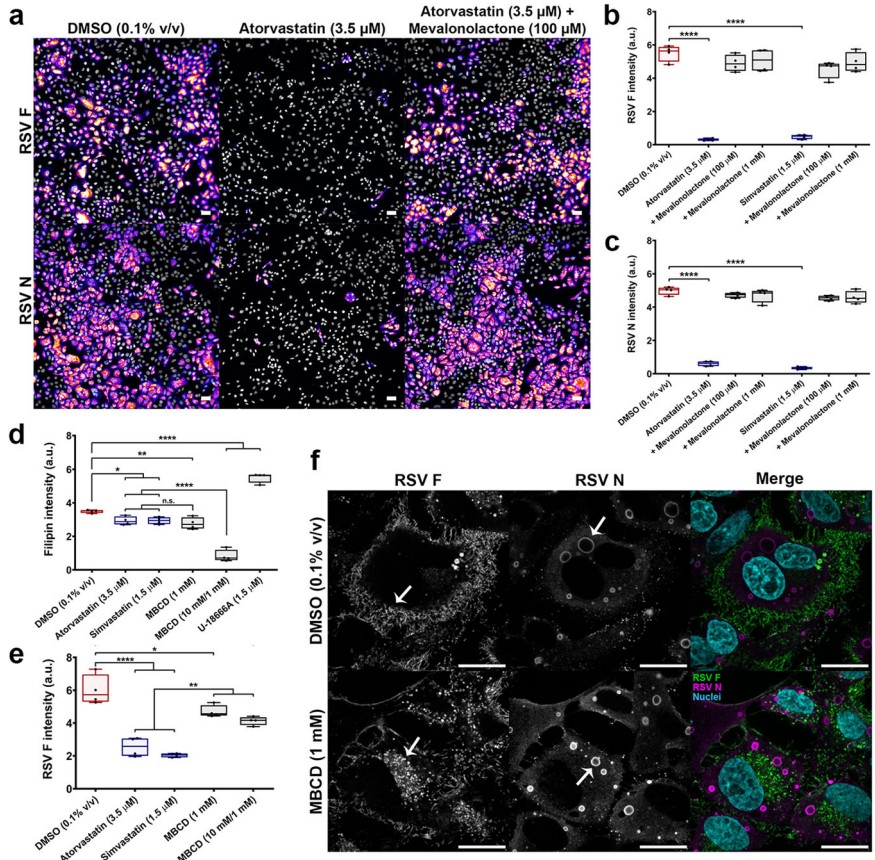

**Fig. 4 The antiviral effects of statins are mediated by the mevalonate pathway.** To confirm that the antiviral effects of statins were due to inhibition of the mevalonate pathway, RSV-infected cells were co-treated 2 h post-viral exposure with atorvastatin and mevalonolactone (a membrane-permeable ester of mevalonate). **a** Representative images of RSV A2 (MOI 0.5)-infected HEp-2 cells acquired 48 h p.i., stained for the F and N proteins. Atorvastatin treatment significantly inhibits RSV, whereas co-treatment with mevalonolactone fully rescues infection. Scale bars = 100 μm. **b, c** Quantification of F and N immunofluorescence 48 h p.i. Exogenous addition of mevalonolactone (100.0 μM or 1.0 mM) restores RSV infection to DMSO baseline in the presence of either atorvastatin (3.5 μM) or simvastatin (1.5 μM) ($n = 4$ independent measurements). **d–f** Mevalonate is converted to several downstream metabolites, including cholesterol. To probe cholesterol-specific effects on RSV infection, HEp-2 cells were depleted of cholesterol using MBCD. **d** Low-dose MBCD (1.0 mM) matched the level of cholesterol depletion by statins, whereas acute pre-treatment with high-dose MBCD (10.0 mM) caused high levels of cholesterol depletion. U-18666A was included as a control and induced intracellular accumulation of cholesterol. Total cholesterol was quantified by filipin III fluorescence ($n = 4$ independent measurements). **e** Both low and high cholesterol depletion by 1.0 and 10.0 mM MBCD had a lesser inhibitory effect on RSV compared to statins, as quantified by F protein intensity ($n = 4$ independent measurements). **f** High-resolution confocal microscopy of the F (green) and N (magenta) proteins in RSV A2-infected HEp-2 cells (MOI 0.5). Cholesterol depletion by MBCD (1.0 mM) disrupts F protein surface localization (as indicated by the arrows). The N protein was expressed and localized to cytoplasmic viral inclusions. Scale bars = 20 μm. Box plots indicate the complete range (whiskers), interquartile range (box), and median (line) of measurements. The y-axis of each graph denotes fluorescent intensity normalized to total nuclear count (a.u. arbitrary units). Statistics were calculated by a one-way ANOVA with a Dunnett's multiple comparisons post hoc test (n.s. non-significant, *$P < 0.05$, **$P < 0.01$, ****$P < 0.0001$).

increased efficiency in screening for viral inhibitors. Here, we have described an automated, high-throughput approach to screen for RSV inhibitors using a re-engineered form of the virus expressing GFP (RSV-GFP). Cells infected with RSV-GFP produce a strong fluorescent signal amenable to high content analysis, as reflected in the Z-factor of $0.65 \pm 0.19$ determined for our assay. Our approach also circumvents the need to perform additional assays to quantify cell viability, as we are able to measure cell death through integrated image analysis of nuclear morphology and staining intensity[22]. This additional step in the analysis pipeline minimizes the risk of identifying false-positive viral inhibitors, which adversely affect host cell health rather than the virus. After screening 2400 FDA-approved and bioactive compounds, we categorized five generalized drug classes capable of inhibiting RSV, including statins, steroid hormones, cardiac glycosides, anti-infectives, and flavonoids. There is evidence in the literature that several of the identified hits exert antiviral effects against

RSV and other viruses[45–48], providing independent validation of our assay as a predictive preclinical tool to screen for viral inhibitors. Based on our findings, we have published a separate report outlining an antiviral mechanism for cardiac glycosides through perturbation of intracellular ion homeostasis[49]. In the present study, we have described our assay methodology and the complete results of our drug screen for the first time. Additionally, we focus on characterizing the effects of cholesterol reduction and protein prenylation on RSV infection.

Virus-induced changes to host cell metabolism have been known to occur for several decades; however, this phenomenon has only recently gained recognition for its essential role in disease pathogenesis[8,50,51]. Notably, a broad range of viruses have been characterized to dysregulate lipid metabolism, including hepatitis C virus[52], Dengue virus[53], and human immunodeficiency virus[54]. Recently, it was found that RSV also causes dramatic changes to the lipidomes of infected tissue in mice[55].

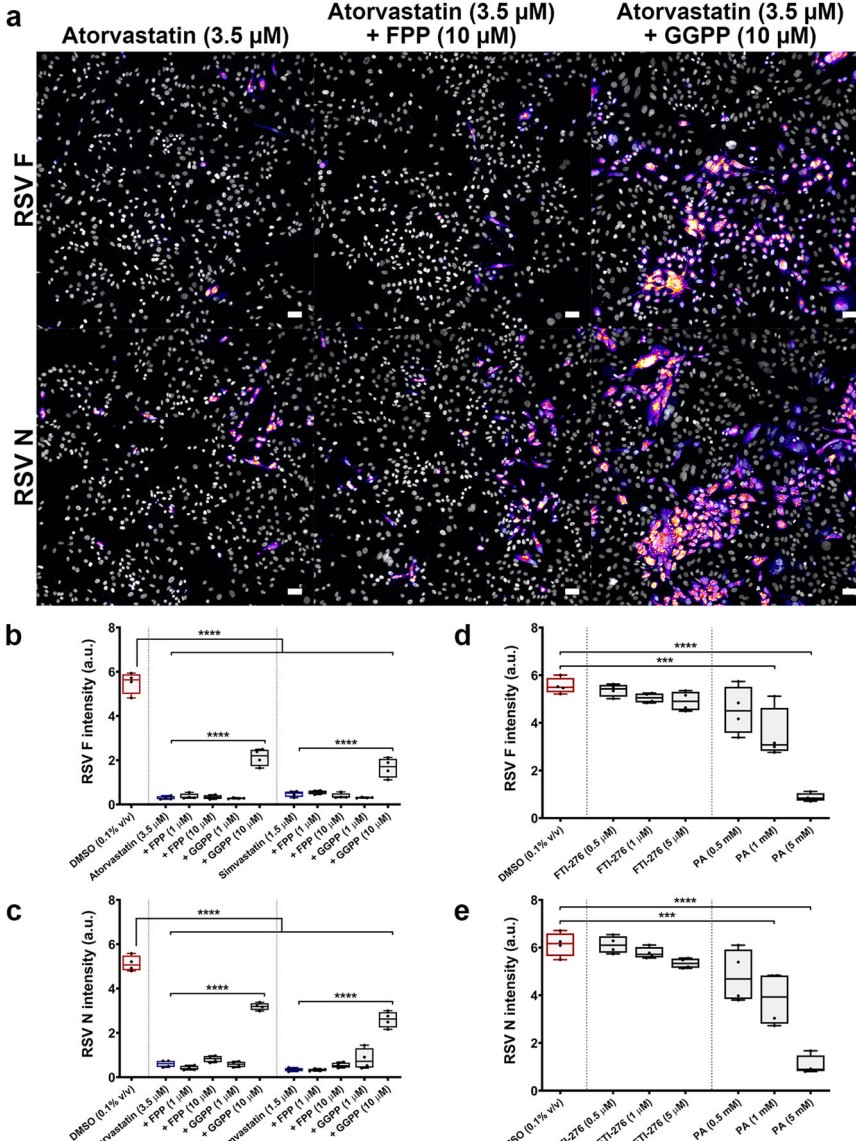

**Fig. 5 Protein geranylgeranylation is specifically required for RSV infection.** RSV-infected cells were co-treated 2 h post-viral exposure with statins and isoprenoids, farnesyl pyrophosphate (FPP) or geranylgeranyl pyrophosphate (GGPP). **a** Representative images of RSV A2-infected HEp-2 cells (MOI 0.5) acquired 48 h p.i., stained for F and N. Atorvastatin (3.5 μM) inhibits the virus, which is partially rescued by co-treatment with GGPP (10.0 μM), but not with FPP (10.0 μM). Scale bars = 100 μm. **b–e** Quantification of F and N immunofluorescence 48 h p.i. **b, c** Exogenous addition of GGPP (10.0 μM) partially rescued the virus in the presence of atorvastatin (3.5 μM) and simvastatin (1.5 μM), whereas FPP did not significantly alter the inhibitory effect of statins ($n = 4$ independent measurements). **d, e** Treatment with the geranylgeranyltransferase inhibitor perillyl alcohol (PA) dose-dependently reduced viral protein intensity, whereas the farnesyltransferase inhibitor FTI-276 did not cause any significant changes ($n = 4$ independent measurements). Both inhibitors were tested at concentrations exceeding their $IC_{50}$. Box plots indicate the complete range (whiskers), interquartile range (box), and median (line) of measurements. The y-axis of each graph denotes protein fluorescent intensity normalized to total nuclear count (a.u. arbitrary units). Statistics were calculated by a one-way ANOVA with a Dunnett's multiple comparisons post hoc test (***$P < 0.001$, ****$P < 0.0001$).

Our results support the concept of lipid dysregulation by RSV, as we observed lipid metabolic pathways to be highly functionally enriched among inhibitors of the virus. Many viruses favor a lipid-rich environment for infection, with certain lipids required for virus uptake, assembly, virion stability, and host immune evasion[50,51]. Several studies have shown that RSV propagation is also reliant on lipid metabolites, namely cholesterol, for these processes[12,13]. The virus has been found to promote biosynthetic activity in lipid metabolic pathways by transcriptional upregulation of lipid metabolic enzymes (e.g. HMGCR)[56]. Lipid metabolites, such as mevalonate, have been measured at increased levels in clinical isolates obtained from RSV-infected infants[57],

indicating that virus-mediated lipid dysregulation is relevant to the human disease. A primary outcome of our screen was identification of the HMGCR antagonists, statins, as inhibitors of RSV infection. We hypothesized that these drugs inhibited RSV by counteracting changes to host lipid metabolism induced by the virus. Accordingly, we found statin-mediated inhibition of RSV was negated by exogenous addition of mevalonate, indicating these drugs exert their antiviral effects through this lipid metabolic pathway. Of related interest, we also found non-statin inhibitors of lipid metabolism to reduce viral infection, including flavonoids[58], celastrols[59], and estrogens[60] (Supplementary Table 1). Our results indicate that RSV promotes a lipid-rich

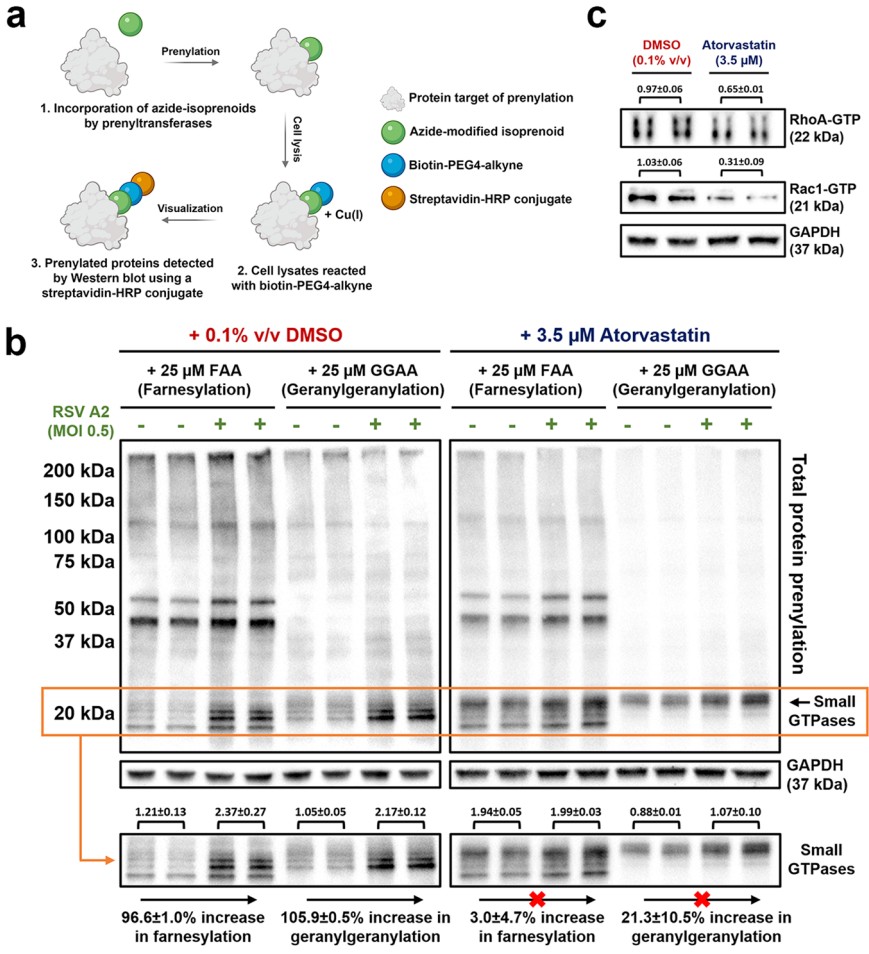

**Fig. 6 Statins mitigate RSV-induced increases to Rho GTPase prenylation and activity. a** A click chemistry-based approach was used to isolate and visualize prenylated proteins. HEp-2 cells were treated with azide-modified isoprenoids, farnesyl alcohol azide (FAA) or geranylgeranyl alcohol azide (GGAA), and infected with RSV A2 (MOI 0.5). Lysates were collected 48 h p.i. and conjugated to biotin-PEG4-alkyne by a copper-catalyzed reaction. Prenylated proteins were then visualized by western blot using a streptavidin-HRP probe. **b** RSV A2 infection of HEp-2 cells (MOI 0.5) induced an increase in total protein prenylation (farnesylation and geranylgeranylation) relative to uninfected cells 48 h p.i. Notably, the virus increased the prenylation of ~20 kDa proteins corresponding to small GTPases (as outlined on the blot). Atorvastatin (3.5 μM) treatment 2 h post-viral exposure inhibited the RSV-induced increase to small GTPase prenylation (both farnesylation and geranylgeranylation). **c** The subcellular localization and activity of Rho GTPases is regulated by prenylation. Active, GTP-bound RhoA and Rac1 levels were measured 48 h p.i. by affinity purification of RSV A2-infected HEp-2 lysates (MOI 0.5) +/− atorvastatin (3.5 μM). Atorvastatin reduced the GTP-bound levels of both RhoA and Rac1 in infected cells. Averaged GAPDH-normalized band intensity is denoted by the numbers above each lane. Values are reported as means ± SEM (n = 2 cell lysates). GAPDH controls were processed on a parallel blot with the same samples and protein load.

environment for efficient propagation, and negative modulators of lipid metabolism may be attractive antiviral candidates.

To better elucidate the antiviral mechanism of statins, we sought to determine the consequences of inhibiting either cholesterol or isoprenoid biosynthesis on RSV infection. Cholesterol is required for the maintenance of lipid raft integrity, membrane structures which are involved in both RSV attachment and assembly[12,61]. Regions of membrane rich in caveolae, a type of lipid raft, are key sites for viral morphogenesis[62]. During assembly, the F protein is targeted to caveolae via residues in the transmembrane domain and the cytoplasmic tail of this protein then acts as a site for the recruitment of other viral components[11,29,30,62]. Interestingly, caveolae ranked highly as an enriched GO cellular component in our network analysis, indicating that several protein targets of hits from our drug screen interact with these structures. Statins, in particular, are known to decrease the structural integrity of caveolae by depleting the membrane of its essential components, cholesterol and Cav3[63]. To mimic the loss of cholesterol induced by statins, independent

of its other cellular actions, we treated RSV-infected cells with MBCD. Consistent with previous studies, our results show that perturbation of lipid raft integrity by cholesterol depletion hinders the localization of the F protein to the plasma membrane. Cholesterol depletion by MBCD, however, was not sufficient to achieve the same overall level of viral inhibition as statins. This indicated that statins inhibit RSV through a combination of both cholesterol-dependent and independent effects.

Mevalonate is converted to other downstream metabolites in addition to cholesterol, including isoprenoids. Statin-mediated inhibition of HMGCR prevents the biosynthesis of isoprenoid substrates required for protein prenylation. We found RSV infection causes an increase in total protein prenylation relative to uninfected cells. This increase in prenylation was mitigated by statin treatment, supporting our hypothesis that statins negate RSV-induced upregulation of host lipid metabolism. Presently, the effects of protein prenylation on viral infection have not been thoroughly explored. However, there is an emerging body of evidence indicating that protein prenylation is usurped by a broad

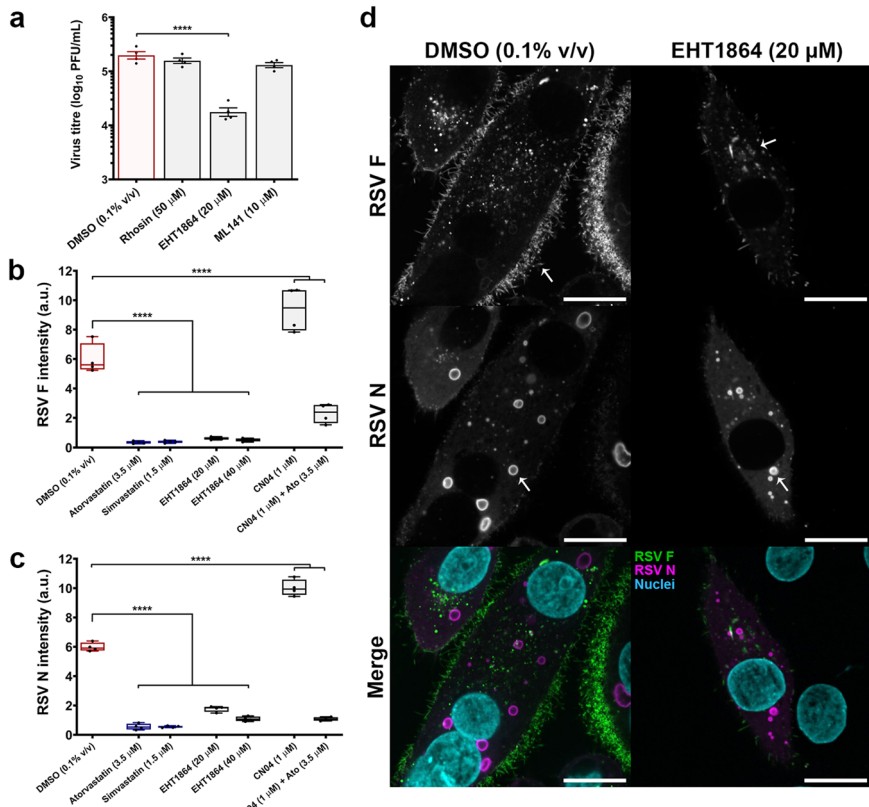

**Fig. 7 Rac1 inhibition disrupts surface expression of the F protein and infectious virus production. a** Infectious virus production was quantified by plaque assay from the supernatants of RSV A2-infected HEp-2 cells treated with inhibitors of RhoA (Rhosin, 50 μM), Rac1 (EHT 1864, 20 μM), or Cdc42 (ML 141, 10 μM). Rac1 inhibition by EHT 1864 (20.0 μM) caused a significant reduction in infectious virus production. Values are reported as means ± SEM ($n = 4$ independent measurements). **b, c** HEp-2 cells were infected with RSV A2 (MOI 0.5) and viral protein intensity was quantified by immunofluorescence 48 h p.i. Rac1 inhibition by EHT 1864 (20.0 μM) caused a significant decrease in F and N protein intensity. Conversely, treatment with a Rho GTPase activator (CN04) enhanced RSV infection and increased protein intensity relative to DMSO control (0.1% v/v). Co-treatment of atorvastatin (3.5 μM) and CN04 (1.0 μM) mitigated the pro-viral effects of Rho GTPase activation. Box plots indicate the complete range (whiskers), interquartile range (box), and median (line) of measurements. The y-axis denotes protein fluorescent intensity normalized to total nuclear count (a.u. arbitrary units) ($n = 4$ independent measurements). **d** High-resolution confocal microscopy of the F (green) and N (magenta) proteins in RSV A2-infected HEp-2 cells (MOI 0.5). Rac1 inhibition by EHT 1864 (20.0 μM) decreases the surface expression of the F protein, and the N protein localizes to cytoplasmic inclusions (as indicated by the arrows). Scale bars = 20 μm. Statistics were calculated by a one-way ANOVA with a Dunnett's multiple comparisons post hoc test (****$P < 0.0001$).

range of viruses[64], highlighting the potential of targeting this process as a host-directed antiviral strategy. A series of promising studies on hepatitis C and D viruses have demonstrated the therapeutic efficacy of prenyltransferase inhibitors[65–68]. A human trial of lonafarnib, an investigational farnesyltransferase inhibitor, saw reduced hepatitis D viral titers following oral administration in chronically infected patients[67]. The results of this study may be applicable to the treatment of other viral infections, including RSV. Geranylgeranylation of RhoA was previously shown to be induced by RSV infection[69]. Concordantly, we demonstrated an important role for geranylgeranylation in our study by rescuing RSV infection through exogenous GGPP addition in the presence of statins, and mitigating infection through geranylgeranyl-transferase inhibition by PA.

Rho GTPases comprise a family of prenylated proteins (RhoA, Rac1, Cdc42) that are key mediators of cytoskeletal organization and host factors for many viruses. Previous studies have found that components of the cytoskeleton, namely actin, are centrally involved in viral transcription and morphogenesis[70–72]. Specific roles have been identified for each Rho GTPase at discrete stages in the replicative cycle of RSV. Cdc42 activity is involved in entry[15], while RhoA activity influences viral filament morphology and cell-to-cell fusion[69,73]. Rac1 was recently described to have a direct role in viral assembly[28], consistent with our observation

that surface expression of the F protein was decreased by EHT 1864-mediated inhibition of the protein. Targeting of viral gly-coproteins to lipid rafts is an essential step in the assembly of virions[29,30]. There is now evidence supporting an additional requirement for the presence and activity of Rac1 in these membrane microdomains for RSV morphogenesis and assembly[28]. Accordingly, expression of a dominant-negative Rac1 mutant was shown to block the development of viral filaments[28]. Localization of Rac1 to the plasma membrane is dependent on post-translational prenylation of the protein, a process which we found to be promoted by viral infection and blocked by statins. A loss in the localization and activity of Rac1 at the plasma membrane likely contributes to the inhibitory effect of statins on RSV assembly. Other small GTPases have also been associated with the replicative cycles of viruses. Specifically, the small GTPase, Rab11, has been implicated in viral protein trafficking and assembly[74,75]. We investigated the effects of modulating Rab11 activity on RSV infection in HEp-2 cells. Inhibition of Rab11-mediated trafficking by the small-molecule CDKI-73[76] caused redistribution of viral proteins within the cell, but did not reduce RSV-GFP infection rates (Supplementary Fig. 4). Though our results highlight a critical role for Rac1 in the replicative cycle of RSV, crosstalk is known to occur between the downstream effectors and signaling pathways of Rho GTPases. It is likely the coordinated actions of

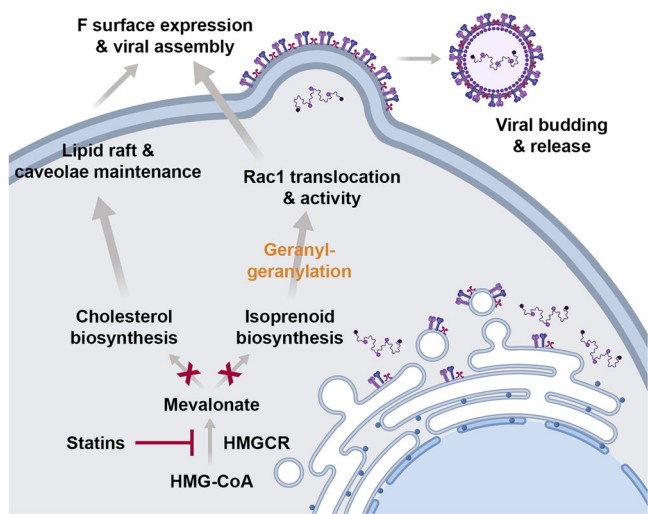

**Fig. 8 A proposed mechanism for statin-mediated inhibition of RSV.**
Statins inhibit HMGCR, thereby preventing the conversion of HMG-CoA to mevalonate. Resultant perturbations to cholesterol and isoprenoid biosynthesis adversely affect the production of infectious virus by the following: (1) Cholesterol is a key component of lipid rafts and caveolae, which are sites for viral assembly. Disruption of lipid rafts reduces the surface localization of viral glycoproteins and inhibits assembly. (2) Isoprenoids are substrates for the prenylation of Rho GTPases, key viral host factors that mediate actin reorganization in the cell. Inhibition of Rac1 prenylation and activity at the plasma membrane disrupts surface expression of the fusion protein and virus morphogenesis.

multiple small GTPases is essential to the virus, underscoring the importance of disrupting virus-induced increases to the prenylation and activity of these proteins.

In summary, our study provides evidence supporting a general mechanism for statin-mediated inhibition of RSV inclusive of its effects on both cholesterol and isoprenoid biosynthesis (Fig. 8). Moving forward, it is important to consider the practical application of lipid modulators, such as statins or geranylgeranyl-transferase inhibitors, as an antiviral therapy. Given that these drugs inhibit the production of infectious virus, they could potentially be given to individuals with an established infection to yield a therapeutic benefit. A number of epidemiological studies have associated statin use with decreased severity of respiratory viral infections and improved clinical outcomes for IAV and SARS-CoV-2[77–81]. Although in vivo data on statins and RSV are limited, a single study found inhibition of RSV in mice by daily treatment with high doses of lovastatin (50 mg/kg) delivered orally[82]. Optimization in the formulation and delivery of the drug could lower the dose to a value more realistic for use in humans. Because RSV targets the respiratory epithelium, aerosolized delivery of statins could maximize the effective concentration at the site of infection while minimizing systemic delivery[83]. Our findings have also highlighted the importance of protein prenylation and Rho GTPase activity for RSV infection. Direct modulation of Rho GTPase geranylgeranylation through the use of prenyltransferase inhibitors, such as PA, is sufficient to inhibit RSV in vitro and warrants further investigation in animal models and humans. Intranasal delivery of PA has been shown to be well-tolerated in humans[84] and could be a feasible treatment for RSV or other respiratory viruses. Overall, our results demonstrate a clear potential for targeting lipid metabolic processes as a host-directed antiviral strategy. Modulation of cholesterol biosynthesis or protein prenylation by statins, prenyltransferase inhibitors, or

other compounds should be explored in future studies to address the growing healthcare burden of RSV.

## Methods

**Cells, viruses, and reagents.** HEp-2 cells are highly permissive to RSV and a common cell model for infection. HEp-2 cells (ATCC; CCL-23) were maintained in a humidified incubator at 37 °C/5% $CO_2$ in Eagle's minimum essential medium (EMEM) (320-005-CL, Wisent Bioproducts) supplemented with 10% v/v fetal bovine serum (FBS) (A3160701, Thermo Fisher Scientific). Cells were infected with either RSV A2 (ATCC; VR-1540) or a recombinant strain of RSV expressing green fluorescent protein (rgRSV224, RSV-GFP) provided by Dr. M. E. Peeples (Children's Research Institute, Columbus, OH) and Dr. P. L. Collins (National Institutes of Health). RSV stocks were prepared as per the referenced protocol[85]. Atorvastatin calcium salt trihydrate (PZ0001), simvastatin (S6196), methyl-β-cyclodextrin (C4555), DL-mevalonolactone (M4667), farnesyl pyrophosphate ammonium salt (F6892), geranylgeranyl pyrophosphate ammonium salt (G6025), FTI-276 tri-fluoroacetate salt (F9553), perillyl alcohol (218391), and biotin-PEG4-alkyne (764213) were purchased from MilliporeSigma. Rhosin (5003), EHT 1864 (3872) and ML 141 (4266) were purchased from Tocris Bioscience. Farnesyl alcohol azide (13269) was purchased from Cayman Chemical. Click-iT geranylgeranyl alcohol azide (C10249) was purchased from Thermo Fisher Scientific. Rho/Rac/Cdc42 Activator I (CN04) was purchased from Cytoskeleton Inc. CDKI-73 (HY-12445) was purchased from MedChemExpress. Goat anti-RSV polyclonal antibody (B65860G) was purchased from Meridian Life Science.

**RSV-GFP infection of HEp-2 cells.** HEp-2 cells were seeded to black, flat, microclear-bottom 96-well plates (655096, Greiner Bio-One) at a density of 5000 cells per well and incubated at 37 °C/5% $CO_2$ overnight. The following morning, cells were washed with serum-free EMEM and treated with RSV-GFP (MOI 0.5) in a minimal volume of serum-free media for 90 min under frequent agitation. Unbound virus was then removed and replaced with fresh serum-containing EMEM. All drug additions were performed 2 h post-viral exposure, unless otherwise stated. Nuclei were stained with Hoechst 33342 (H3570, Thermo Fisher Scientific) (at a final concentration of 1 µg/mL) 48 h p.i. and imaging was performed using the Cellomics ArrayScan VTI HCS Reader at ×10 magnification (SPARC BioCentre, The Hospital for Sick Children, Toronto, ON).

**Automated image analysis of RSV-GFP infected cells.** Automated quantification of infection rates from fluorescent images were performed in CellProfiler (v2.2.0)[86]. Major steps in the image analysis pipeline included: (1) background subtraction and smoothing, (2) segmentation of nuclei, (3) segmentation of GFP-fluorescent area, (4) object masking, and (5) measurement of nuclear number, size, and intensity. Infection rate was quantified from the ratio of nuclei within GFP-fluorescent area to the total number of nuclei (Fig. 1b). This method of analysis accounts for in vitro syncytium formation induced by RSV, whereby a single infected cell may arise from the fusion of multiple cells. Mean nuclear fluorescent intensity and size was quantified as a measure of cell viability[22].

**RSV inhibitor screen.** The MicroSource SPECTRUM Collection (Discovery Systems Inc.) consisting of 2400 compounds was screened using the previously outlined RSV-GFP infection assay protocol. Robot-assisted addition of drugs was performed within 1 h of RSV-GFP infection and infection rates were assayed 48 h p.i. Drugs were provided as 10 mM stocks in DMSO and serial diluted to a final concentration of 10 µM, resulting in a final DMSO concentration of 0.1% v/v. The nucleolin aptamer AS1411 (80 µM), which has previously been shown to inhibit RSV infection in preclinical models[87], was used as a positive drug control and DMSO (0.1% v/v) served as a vehicle control. Hits were selected based on the criteria that: (a) the tested drug caused a >50% reduction in infection rate and (b) was not overtly cytotoxic (<10% change in mean nuclear intensity relative to DMSO control). Z-factors were calculated on a per-plate basis to monitor assay quality[88]. Selected hits were then validated over a range of concentrations, and this data were used to generate dose–response curves for each drug in HEp-2 cells (Supplementary Fig. 1).

**GO enrichment analysis of biological processes and cellular components.** To determine host pathways utilized by RSV during infection, protein targets corresponding to each drug screen hit were identified using DrugBank (https://www.drugbank.ca/). Out of 60 drug screen hits, 31 compounds had known protein targets which generated a final list of 133 unique proteins (Supplementary Table 2). These proteins were queried in PANTHER for GO functional enrichment of biological pathways and cellular components using the default settings. Visual interactome analysis was performed using the ClueGO plugin (v2.5.5) in Cytoscape (v3.7.2)[89,90].

**In vitro cytotoxicity assay.** Quantification of nuclear size and intensity was used for preliminary assessment of cellular viability. To confirm that viral inhibition was

not due to drug-related cytotoxicity when validating the activity of statins, cell viability was assayed using the MTT Cell Proliferation Kit I (11465007001, Roche) as per the manufacturer's protocol.

**Well-differentiated primary nasal epithelial cultures (WD-PNECs)**. WD-PNECs were cultured as per the referenced protocol[49]. Using a sterile cytology brush, nasal scrapings were obtained from healthy volunteers with no known history of lung disease or respiratory medication use. Cells were then seeded on a collagen-coated flask and maintained in Bronchial Epithelial Cell Growth Medium (BEGM) (CC-3170, Lonza) or PneumaCult-Ex (05008, STEMCELL Technologies). Upon reaching 70% confluency, cells were passaged and seeded on collagen-coated transwell inserts (6.5 mm diameter, 0.4 μm pore size) (3413, Corning). An air–liquid interface was generated by removing media on the apical side of the cells and replacing the remaining media with PneumaCult-ALI media (05001, STEM-CELL Technologies). By week 3, cells were differentiated with a ciliated phenotype. Transepithelial resistance was quantified using an ohmmeter (World Precision Instruments) to evaluate epithelial barrier function prior to experimentation. Cells were apically infected with RSV-GFP (MOI 0.01) and infection rates were quantified from GFP signal.

**Immunofluorescence staining and microscopy of RSV F and N**. HEp-2 cells were seeded to No. 1 glass coverslips (1014355117NR1, Thermo Fisher Scientific) for high-resolution confocal microscopy, or black, flat, microclear-bottom 96-well plates (655096, Greiner Bio-One) for high-throughput quantification of RSV F and N immunofluorescence. Cells were infected with RSV A2 (MOI 0.5). At 48 h p.i., cells were fixed in 4% paraformaldehyde for 30 min and permeabilized with 0.3% Triton X-100 for 10 min at room temperature. F and N proteins were probed using mouse anti-RSV F (RSV3216) and mouse anti-RSV N (RSV3132) mAbs (Bio-Rad Antibodies) diluted 1:500 in PBS + 0.1% Tween-20 (PBST), or mouse anti-RSV F mAb Alexa Fluor 488 conjugate (Millipore Sigma, 133/1H) diluted 1:200 in PBST over a 1-h incubation at room temperature. This was followed by detection with secondary anti-mouse Fab fragments conjugated to Alexa Fluor 488 (4408) or 647 (4410) (Cell Signaling Technology) diluted 1:1000 in PBST over a 1-h incubation at room temperature. Fluorescent images were acquired using either the Cellomics ArrayScan VTI HCS Reader at ×10 magnification (for quantification) or Olympus IX81 Spinning Disk Confocal microscope at ×60 magnification with oil immersion (for high-resolution images). F and N immunofluorescence was quantified in CellProfiler and normalized to total nuclear count.

**Time-of-drug addition assay**. Time-of-drug addition assays are routinely used to assess the approximate stage of viral infection affected by a drug[25]. HEp-2 cells were seeded to black, flat, microclear-bottom 96-well plates (Greiner Bio-One) and infected with RSV-GFP (MOI 2.0). Atorvastatin addition was performed 2 h pre-infection and 2, 4, 6, 8, 16, and 22 h post-infection. RSV-GFP infection rates were quantified 24 h p.i to approximately capture a single cycle of infection.

**Quantification of infectious virus production**. HEp-2 cells were seeded to six-well plates at a density of $1 \times 10^6$ cells per well and incubated overnight at 37 °C/5% $CO_2$. The following morning, cells were infected with RSV A2 (MOI 1.0) and treated with drugs 2 h post-viral exposure. The supernatant, containing progeny virions, was collected 48 h p.i., snap frozen, and stored at −80 °C. Quantification of infectious virus particles was performed by a traditional plaque assay[91]. HEp-2 cells were seeded to six-well plates at a density of $1.5 \times 10^6$ cells per well and grown to 90% confluency. Cells were incubated with serial dilutions of the collected supernatants for 90 min at 37 °C/5% $CO_2$ under frequent agitation. The inoculant was then removed and replaced with 4 mL of 1:1 4% FBS DMEM-F12/1% agarose. Plates were incubated for 5 days at 37 °C/5% $CO_2$. Cells were then fixed with 10% formalin for 30 min at room temperature. After fixation and removal of agarose from each well, the cells were stained with 0.05% neutral red and plaques were quantified under a dissecting microscope.

**Cholesterol depletion and quantification**. Cellular cholesterol was quantified by filipin III fluorescence using the Cholesterol Cell-Based Detection Assay Kit (Cayman Chemical) as per the manufacturer's protocol. Cholesterol depletion was performed using methyl-β-cyclodextrin (MBCD), a water-soluble oligosaccharide that binds cholesterol at the plasma membrane[92]. Treatment with 1 mM MBCD for 48 h resulted in a moderate reduction of total cellular cholesterol in HEp-2 cells, comparable to statin treatment. Addition of a 30-min pre-treatment step with 10 mM MBCD resulted in a high reduction of total cellular cholesterol without causing cell death. Fluorescent images were quantified in CellProfiler (v2.2.0).

**Measurement of protein prenylation by click chemistry**. HEp-2 cells were cultured in the presence of farnesyl alcohol azide (FAA) (SC-294586, Santa Cruz Biotechnology) or geranylgeranyl alcohol azide (GGAA) (C10249, Thermo Fisher Scientific) at a final concentration of 25 μM diluted in 10 mM aqueous $NH_4OH$. Cells were infected with RSV A2 (MOI 0.5) and treated with drugs 2 h post-viral exposure. Cell lysis was performed on ice in RIPA buffer 48 h p.i., and prenylated proteins were reacted with biotin-PEG4-alkyne using the Click-iT Protein Reaction

Buffer Kit (C10276, Thermo Fisher Scientific) as per the manufacturer's protocol. Prenylated proteins in the processed samples were then visualized by western blot using a streptavidin-HRP conjugate (RPN1231, Cytiva) and Clarity ECL Substrate (1705061, Bio-Rad). Relative protein levels were quantified by densitometry in ImageJ (v1.52b).

**Active RhoA/Rac1 affinity purification assays**. HEp-2 cells were infected with RSV A2 (MOI 0.5) and treated with drugs 2 h post-viral exposure. Cell lysis was performed on ice in RIPA buffer 48 h p.i. and GTP-bound RhoA or Rac1 were detected using the Active Rho Detection Kit (8820) and Active Rac1 Detection Kit (8815) (Cell Signaling Technology) as per the manufacturer's protocols. Proteins were visualized by western blot using the reagents and antibodies included with each kit. Quantification was performed by densitometry in ImageJ (v1.52b).

**Statistics and reproducibility**. Statistical analyses were performed in GraphPad Prism (v6.0) (GraphPad Software) using a one-way ANOVA with a Dunnett's multiple comparisons post hoc test. All values within the manuscript are presented as means ± SEM. A description of replicates is provided in each figure caption. Box plots indicate the complete range (whiskers), interquartile range (box), and median (line) of measurements. Significant differences between groups of data are represented as n.s. non-significant, $*P < 0.05$, $**P < 0.01$, $***P < 0.001$, and $****P < 0.0001$.

**Reporting summary**. Further information on research design is available in the Nature Research Reporting Summary linked to this article.

## Data availability
The authors declare that the data supporting the findings of this study are available within the paper and its supplementary information files. Detailed data for each graph can be found in Supplementary Data 1. Uncropped western blots can be found in Supplementary Fig. 5.

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

## Acknowledgements

The authors wish to thank SPARC BioCentre and The Imaging Facility at The Hospital for Sick Children for use of robotics and imaging equipment. We also would like to thank Dr. M. E. Peeples and Dr. P. Collins for kindly providing RSV-GFP and Dr. R. Hegele for providing AS1411. Graphical representations were created with BioRender.com. J.T.M. would like to thank the Department of Anesthesia and Pain Medicine (The Hospital for Sick Children) for protected research time, the Wasser Family and SickKids Foundation as the holder of the Wasser Chair in Anesthesia and Pain Medicine, and the University of Toronto Department of Anesthesiology and Pain Medicine as the holder of a Merit Award.

## Author contributions

M.M. conceptualized and performed all experiments, analyzed the data, and wrote the manuscript. W.D. assisted with the drug screen. M.J.N. and W.D. propagated virus, cultured well-differentiated primary human nasal epithelial cells, and edited the manuscript. T.J.M. aided in the conceptualization of experiments, edited the manuscript, and supported the project. J.T.M. supervised all experiments, edited the manuscript, and provided financial support.

## Competing interests

The authors declare no competing interests.
