## [Transparent Peer Review File · Communications Biology]

Response to Reviewers

Reviewer #1 (Remarks to the Author):

In this manuscript, the authors report a live cell-based high content screening approach and identified five classes of drugs, statins, steroid hormones, cardiac glycosides, anti-infectives and flavonoids, which inhibited RSV infection. Among them, they further analyzed and validated the effects of statins on RSV infection in HEP-2 and primary human nasal epithelial cells (PNECs) grown on an air-liquid interface. They showed that statin-mediated inhibition of RSV was negated by exogenous addition of mevalonate, indicating that statins exert their antiviral effects through lipid metabolic pathway. The authors also showed that exogenous addition of GGPP restored F and N expression, suggesting that a loss in protein prenylation contributed to statin antiviral activity. Using specific inhibitors, they suggested that disruption of prenylation of Rac1 inhibited the virus through a reduction in RSV F expression and surface localization.

Comments and responses:

1. Because of significant burden of RSV infection, development of targeted therapies against the virus is imperative. However, statins that they analyzed in this manuscript are already known to inhibit RSV and other respiratory RNA viruses, and therefore, this report itself is not novel.

The reviewer correctly notes that previous studies have identified statins as inhibitors of RSV and other viruses, a point which we have referenced in our manuscript. The novel aspects of our study pertain to: (a) the development and application of a high content screen for rapid identification of RSV inhibitors (which also yielded other novel inhibitors of RSV), and (b) highlighting the importance of host lipid metabolites, specifically isoprenoids, to RSV infection. Reports of statin-mediated antiviral activity have largely focused on the effects of depleting cellular cholesterol and lipid raft disruption. Our study is the first to show that RSV globally upregulates host protein prenylation, and statins counteract these virus-induced changes to inhibit infection. Furthermore, we have provided evidence towards a specific requirement of geranylgeranylation for RSV infection. Modulation of protein prenylation is an unexplored treatment paradigm in the context of RSV. To demonstrate the therapeutic potential of targeting prenylation, we showed that perillyl alcohol (a geranylgeranyl transferase inhibitor with favorable pharmacokinetic profiles in humans) can significantly inhibit RSV infection *in vitro*.

2. Also, their analysis on the mechanism of inhibition is incomplete, and there are some issues, especially on experimental conditions. Most of the experiments were done with cells infected at MOI of 0.5 and analyzed at 48 h post infection, at which time viruses already spread from initially infected cells. This condition confuses the interpretation of the inhibitory effect on viral replication/protein production. Also, viral protein (F and N) expression was shown as intensity of the IF signal/cell, which makes the readout complexed. These assays should be done at earlier time points by Western blot analysis if the authors intend to determine the effect of statins on viral replication (protein production). Additional approach such as quantification of viral mRNAs would be informative to unveil the steps statins inhibit viral infection.

We thank the reviewer for their suggestions. To address the highlighted concerns, we have performed the following additional experiments:

(1) a time-of-drug addition assay with an earlier endpoint of 24 hours to capture a single cycle of infection, at MOI 2.0 (instead of MOI 0.5) (Panel A), (2) quantification of infectious virus

production by plaque assays on supernatants collected from RSV A2-infected HEp-2 cells +/- statin treatment (Panel B), and (3) Western blot analysis of protein levels in lysates collected 24 h p.i. from RSV A2-infected HEp-2 cells +/- statin treatment (Panel C, also see Supplementary Figure 2). We have also revisited our data interpretation and conclusions in the manuscript (changes are highlighted throughout the main text).

When performing the time-of-drug addition assay, our primary goal was to ascertain whether entry events or post-entry events in the replicative cycle of RSV were targeted by statins. Our results consistently showed that statins target post-entry events. This was initially evidenced by two observations: (a) there was no significant difference in the antiviral effect of statins when the drug was added 2 hours pre-infection vs. 2-6 hours post-infection, and (b) disruption to the surface expression of the F glycoprotein, as visualized by immunofluorescence. Previous studies have demonstrated that surface expression of the F protein is an essential step in viral assembly, as the cytoplasmic tail of this protein acts as a site for the recruitment of other viral components^{1,2}. The F protein is known to be targeted to lipid rafts¹, with recent data indicating additional specificity for membrane microdomains containing Rac1³. We have shown lipid raft disruption by MBCD-mediated cholesterol depletion and Rac1 inhibition by EHT 1864 perturbs surface expression of the F protein (Figure 4f, 7d), providing evidence that statins inhibit viral assembly by these mechanisms. To verify this, we used a plaque assay to quantify infectious virus production from the supernatants of statin-treated RSV A2-infected HEp-2 cells (as suggested by both reviewers). Atorvastatin (3.5 µM) and simvastatin (1.5 µM) caused a significant decrease (~10-fold) in infectious virion production. We also probed viral protein expression 24 h p.i. by Western blot (as suggested by Reviewer #1) using a pan-RSV polyclonal antibody which reacts with all viral

antigens. Our results showed that statin treatment caused a decrease in total viral protein levels. Studies on IAV have similarly observed statins to suppress viral protein expression, although the mechanism by which this occurs is not fully understood (e.g. may be due to disruption of lipid droplets and replication hubs, Rac1 inhibition and its effects on viral polymerase complex activity, and/or other factors)^{4,5}. Given the general reliance of viruses on host lipid metabolism, it is not surprising that statins would affect multiple processes within the replicative cycle of RSV.

References:

1. Oomens, A.G.P. et al. (2006). The Cytoplasmic Tail of the Human Respiratory Syncytial Virus F Protein Plays Critical Roles in Cellular Localization of the F Protein and Infectious Progeny Production. *J Virol.* **80(21): 10465–10477.**
2. Batonick, M. & Wertz, G.W. (2011). Requirements for Human Respiratory Syncytial Virus Glycoproteins in Assembly and Egress from Infected Cells. *Adv Virol.* **2011: 343408.**
3. Ravi, L.I. (2021). Virus-induced activation of the rac1 protein at the site of respiratory syncytial virus assembly is a requirement for virus particle assembly on infected cells. *Virology* **557:86-99.**
4. Episcopio, D. et al. (2019). Atorvastatin restricts the ability of influenza virus to generate lipid droplets and severely suppresses the replication of the virus. *FASEB J.* **33(8): 9516–9525.**
5. Dierkes, R. et al. (2014). The Rac1 Inhibitor NSC23766 Exerts Anti-Influenza Virus Properties by Affecting the Viral Polymerase Complex Activity. *PLoS One.* **9(2): e88520.**

3. Also, analysis on the effect of statins on prenylation of small GTPase is incomplete. Other proteins such as Rab GTPases could be the target of the statins, affecting virus assembly and F surface expression. Because their results suggest statins' effect at late time point after virus infection, further analysis on virus assembly are encouraged.

Rab11 is most commonly associated with egress and assembly⁶, viral processes which are affected by statins. The small molecule CDKI-73 was recently described as an inhibitor of Rab11-mediated trafficking⁷. HEp-2 cells were infected with RSV A2 or RSV-GFP (MOI 0.5) and treated 2 hours post-viral exposure with CDKI-73 (0.5 μ M or 1.0 μ M), concentrations at which the drug has been shown to inhibit Rab11 activity *in vitro*⁷. Although Rab11 inhibition caused redistribution of viral proteins in the cell (relative to DMSO control) (Panel A), it did not reduce infection rate in our RSV-GFP assay (Panel B). We have added this information to the manuscript as Supplementary Figure 4.

We performed an additional experiment to quantify infectious virus production via plaque assay from the supernatants of RSV A2-infected HEp-2 cells treated with RhoA, Rac1, and Cdc42 inhibitors. Rac1 inhibition by EHT 1864 significantly decreased infectious virus production (data included in Figure 7a) (Panel C below). We also probed changes to the expression of viral proteins 24 h p.i. by Western blot on RSV A2-infected HEp-2 cell lysates (MOI 0.5), and found Rac1 inhibition caused a decrease in viral protein levels (Supplementary Figure 2). Collectively, our data highlights a central role for Rac1 activity in the replicative cycle of RSV. Inhibition of Rac1 decreases viral protein levels by Western blot (Supplementary Figure 2), reduces surface expression of F (Figure 7d), and inhibits the production of infectious virus (Figure 7a). Our findings are further supported by a recent study that described a direct role for Rac1 in viral assembly³.

References:

3. Ravi, L.I. (2021). Virus-induced activation of the rac1 protein at the site of respiratory syncytial virus assembly is a requirement for virus particle assembly on infected cells. *Virology* 557:86-99.
6. Spearman, P. (2018). Viral interactions with host cell Rab GTPases. *Small GTPases* 9(1-2): 192–201.
7. Sorvina, A. et al. (2020). CDKI-73 is a Novel Pharmacological Inhibitor of Rab11 Cargo Delivery and Innate Immune Secretion. *Cells* 9(2):372.

4. Figure 3A: This figure is misleading. Although macropinocytosis is one of the route of entry for RSV, majority of the virus is considered to fuse at the plasma membrane, which is not shown here. Also, data source of approximate time corresponding to major events should be shown.

We agree with the reviewer that the multiple modes of entry for RSV should have been reflected in our original figure. In our revised manuscript, we have opted to remove this diagram and have instead cited studies in HEp-2 cells that have correlated viral processes to specific times post-infection (see Page 8, Figure 3).

5. Figure 3: it is unclear why a time-of-drug addition assay was conducted with the MOI 0.5 conditions. The readout is 48 h pi. Because positive cells were identified as GFP expression at the stage of protein expression, this condition is not appropriate to address the step statins inhibit the virus. RSV also fuses neighboring cells, and not necessarily forms virions to spread neighboring cells.

The reviewer is correct in noting that time-of-drug addition assays are typically performed with a higher viral inoculum at an earlier endpoint to capture a single cycle of infection. As discussed in our response to ‘Comment #2’, we have reperformed this assay with RSV-GFP (MOI 2.0) and a 24-hour endpoint. RSV-GFP expression can be used to generally inform on whether entry vs. post-entry events are affected by a drug. More targeted experiments (e.g. immunofluorescence of viral proteins, plaque assays, and Western blots) can then be performed to further probe the actions of a drug. Specifically, we demonstrated that statins decrease infectious virus production by plaque assay and do not simply affect viral spread via syncytial formation.

6. Fig. 3C: Unlike F, expression of N seems to be unaffected by Atovastatin, suggesting that the inhibitor does not block viral replication or protein expression, but inhibit surface expression of the glycoprotein. These viral proteins should be quantitated by western blot to conclude a perturbation in viral protein expression.

As discussed in our response to ‘Comment #2’, we have performed Western blot analysis of lysates collected from RSV A2-infected HEp-2 cells 24 h p.i. Statin treatment was verified to reduce viral protein expression. N protein levels were observed to decrease with statin treatment, however, there was a lower relative change to the expression of this protein when compared to other viral proteins.

7. Figure 4E: The authors state that “atorvastatin (3.5 μ M) reduced infection rate by 57.5 \pm 11.4%” in the text, but this is not accurate. RSV F intensity does not correlate with viral infection as seen in Fig. 3C; F negative cells express N, meaning that they were infected.

We have clarified our wording to state changes to “viral protein intensity” or “viral protein levels” rather than “infection rate” when F or N were quantified. These changes are highlighted throughout the revised manuscript.

8. Figure 5DE and text “FTI-276, an inhibitor of farnesyltransferase, did not cause any significant changes to infection rate”. This could be totally dose dependent. Did the authors confirm that the FTI-276 inhibited the farnesyltransferase activity at the concentrations tested in this assay?

Reported values for FTI-276 IC₅₀ are typically in the low nanomolar range. We tested the highest subtoxic concentrations possible (low micromolar) to ensure that the observed lack of efficacy was not dose-dependent. We have also added a statement and citation in the manuscript (Page 12) to clarify this point for readers.

9. Figure 6B. The detected bands were labeled as prenylated small GTPases, but how did the authors identify these bands as “small GTPases”?

Several bioinformatic and practical studies have characterized the eukaryotic prenylome, and small GTPases are the only group of proteins known to be prenylated at this molecular weight (~20 kDa)^{8,9}. For the purposes of our study, we had initially probed the same samples for Rac1 and confirmed the presence of this protein at the corresponding molecular weight. We have added this information to the manuscript text (Page 12) and have included the Western blots for Rac1 in Supplementary Figure 5.

References:

8. Maurer-Stroh, S. et al. (2007). Towards Complete Sets of Farnesylated and Geranylgeranylated Proteins. *PLoS Comput Biol.* 3(4):e66.

9. Nguyen, U. T. T. et al. (2009). Analysis of the eukaryotic prenylome by isoprenoid affinity tagging. *Nat Chem Biol.* 5(4):227-35.

Reviewer #2 (Remarks to the Author):

The authors have developed a nice automated high throughput system for specific screening of drugs against RSV. The authors have used this screening system to identify candidate anti-RSV compounds and then validated the observed effects by inhibitor specific investigations. Finally, they have analysed the mechanistic basis for the inhibition.

This is a well written, exciting and important piece of work given the current lack of general, effective therapeutic strategies for RSV disease.

There are however, several areas that need improvement. These are mostly associated with the use of assays and interpretation of the outcomes.

Comments and responses:

1. Why have the authors not used plaque assays or RT-qPCR anywhere in the manuscript to demonstrate infectious virus production and replication? Currently, all infection assays in the manuscript show the expression of specific viral proteins in infected cells. What is the evidence to show that there is a change in the level of infectious virus produced or released from the cell? This is an important aspect as authors suggest that a later stage in RSV assembly is impacted by

inhibition of Rho prenylation. The current use of immunofluorescence only does not support the authors' major conclusion. Authors correctly state that their data shows that the various inhibitors reduce infection rate; it does not show that there is less infectious RSV produced. The only data showing that statins impact a later step in RSV assembly is shown in Figure 3B. It is not clear at what time after infection were the inhibitors added in the experiments represented in figures 4-7; the assumption is, immediately after infection. Authors should provide data where Rho prenylation inhibition late in RSV infection results in the same outcomes as that shown in figures 4-7.

We thank the reviewer for their kind comments and insightful feedback. Both reviewers rightly suggested that additional assays were needed to support our conclusions. We have more thoroughly addressed the concerns outlined here by Reviewer 2 in our response to 'Comment #2' for Reviewer 1. A reduction in the release of infectious virus was verified by plaque assay on supernatants collected from RSV A2-infected HEp-2 cells treated with statins or the Rac1 inhibitor EHT 1864 (Figure 3b, 7a). Additionally, we performed Western blot analysis on cell lysates collected 24 h p.i. and demonstrated a reduction in viral protein levels by statins and Rac1 inhibition (Supplementary Figure 2). Our findings are supported by prior studies that have demonstrated the importance of F surface expression for viral assembly^{1,2} and the involvement of Rac1 in this process³. To address the reviewer's secondary concern, all drug treatments were performed shortly after incubating cells with the virus for 90 minutes (i.e. post-infection). We have added a clarifying statement to our text and figures that 'drug additions were performed 2 hours post-viral exposure' (highlighted throughout the manuscript).

References:

1. Oomens, A.G.P. et al. (2006). The Cytoplasmic Tail of the Human Respiratory Syncytial Virus F Protein Plays Critical Roles in Cellular Localization of the F Protein and Infectious Progeny Production. *J Virol.* **80(21): 10465–10477.**
2. Batonick, M. & Wertz, G.W. (2011). Requirements for Human Respiratory Syncytial Virus Glycoproteins in Assembly and Egress from Infected Cells. *Adv Virol.* **2011: 343408.**
3. Ravi, L.I. (2021). Virus-induced activation of the rac1 protein at the site of respiratory syncytial virus assembly is a requirement for virus particle assembly on infected cells. *Virology* **557:86-99.**

2. If, as the authors state, initial steps in RSV infection are not affected by statins, we would expect the same number of fluorescent foci (e.g. figure 2A) but fewer syncytia. Can the authors explain how their observation of fewer foci correlates with their conclusion?

The fluorescent images presented in the manuscript were acquired 48 h p.i., at which time secondary infection can occur (spread from initially infected cells). Because statins inhibit the production and spread of virus, you would expect to see fewer foci at this later timepoint relative to untreated cells. However, we do observe a similar number of foci between DMSO control and statin treated cells prior to secondary rounds of infection, indicative that statins are not affecting viral entry.

3. It is very difficult to identify the various cellular compartments in the confocal images. Can the authors please provide images with markers of plasma membrane in all confocal data?

We have added additional markers to our confocal images to clarify the location of the plasma membrane.

4. The images provided suggest that inhibitor treatment results in reduced overall size of the cell, hence the authors' interpretation that the inhibitors used lead to increased accumulation of N protein is not correct. The higher intensity may be the same amount of N concentrated to a smaller area.

We agree with the point raised by the reviewer and have removed all previous statements related to the accumulation of N protein. In the revised manuscript, we only comment on the localization of this protein.

5. In several figures, 'filaments' are indicated and mentioned in the legends. These are not clearly shown. Authors should provide zoomed in images clearly showing the filaments. In addition, currently it is not possible to determine if the staining indicated is within or external to the cell.

The word 'filament' was improperly used at certain points in the initial manuscript draft when referring to F protein projections at the cell surface, causing confusion for the reader. We have corrected all instances of this mistake in the revised manuscript. For reference, the confocal images represent a cross section through the cell (hence the appearance of cytoplasmic viral inclusions as rings).

6. Western blots are provided in figure 6. These would be greatly improved by providing a blot probed for RSV proteins. On an aside, why not present blots as black bands on white background as is usual?

We have included additional Western blots probed for RSV proteins (Supplementary Figure 2), as discussed in our response to Reviewer #1 'Comment 2'. Furthermore, we have changed the presentation of the blots to show dark bands on a light background (as per the literature standard).

7. The y-axis in several figures does not correlate with the description in the text (e.g. 0.8 rather than 80%). Please check and revise all figures accordingly. In many figures, the labelling of the y-axis appears not quite right, e.g. if the units are relative to a baseline, this should be stated. Also, the process of calculation of the these units is not clearly described anywhere.

For Supplementary Figure 1, values were not converted from fractions to percent (i.e. 0.8 vs. 80%) in the initial manuscript draft. We have corrected this mistake and double checked all figures to ensure the axes correspond to the values or text. We have also provided additional clarification on the calculation or normalization of values in each figure caption.

8. Please check the text in page 8 - there are several sentences that appear to be stating the opposite of what is shown in the figures.

We have revisited our interpretation of the data presented on Page 8. Furthermore, revisions have been made in the text to reflect the results of our plaque assay and Western blot experiments.

9. The authors end the discussion with suggestions of the use of the chosen inhibitors in the clinic. This would be made stronger if the authors could clearly show that the inhibitors are effective when given later in infection.

Clinical studies on viral infection and statins or prenyltransferase inhibitors have demonstrated these drugs have a therapeutic effect in patients. Statin use in adults has been shown to decrease the risk of mortality and reduce the overall severity of influenza and, more recently, COVID-19 infection. One caveat of these findings is that most patients were already receiving statins at the time of infection. Prenyltransferase inhibitors have been shown to impart an antiviral effect and improved clinical outcomes in patients with chronic hepatitis infection (i.e. at a later stage of infection). With our *in vitro* model, we have shown that statins given up to 24 h p.i. can inhibit RSV at a 48-hour endpoint. To best determine the efficacy of statins against established infections, however, it would be more appropriate to do a dosing experiment on an *in vivo* model as part of a future study.

Response to Reviewers

Reviewer #1 (Remarks to the Author):

This is a revised manuscript reporting the inhibitory effect of statins and a geranylgeranyl transferase inhibitor on RSV infection. The authors appropriately addressed most of the previous comments, and the manuscript was significantly improved. It includes several important novel data, including a specific requirement of geranylgeranylation for RSV infection. The data presented in this revised manuscript are convincing and justified well for the conclusion. It is an appropriate topic and of interest of general readers of Communication Biology. I have no issues on this revised manuscript.

We would like to thank the reviewer for their comments and thoughtful suggestions to improve the manuscript during the peer review process.

Reviewer #2 (Remarks to the Author):

The authors have made a commendable effort to address all the reviewers' comments, including providing more data and modifying the text.

However, one of my comments is not completely addressed and I have a comment on their Response.

Comments and responses:

1. In their Response, the authors state that they have added additional markers to the confocal images to clarify the location of the plasma membrane. I cannot see this in the revised manuscript, except for a label of PM which appears to be arbitrary. Please provide the staining pattern obtained with the markers you have used (this can be as a supplementary figure) as evidence.

To address the reviewer's comment, we have performed an additional experiment in RSV A2-infected HEP-2 cells (MOI 0.5) to co-stain for viral proteins and the plasma membrane using wheat germ agglutinin (WGA). Representative images of DMSO (0.1% v/v) and atorvastatin (3.5 μ M)-treated cells are included below and in Supplementary Figure 3. It is important to note that WGA also stains glycoproteins, therefore there is overlap in staining with the F protein. Consistent with our previous observations, atorvastatin treatment was associated with a reduction in F protein surface expression.

2. In their Response, the authors state that the nature of the confocal images leads to the ring like appearance of the cytoplasmic inclusions. This is not correct. The ring like appearance suggests the localisation of the protein they are probing for. This seems to suggest a lack of understanding of immunofluorescence assays, which I am sure is not the case.

RSV inclusion bodies have been described in the literature as spherical, liquid-liquid phase separated structures¹⁰. Immunostaining of the N protein, which localizes to viral inclusions, shows that the protein is most concentrated at the boundaries of these spherical structures. Therefore, when observing a Z-stack that cuts through the centre of the inclusion body, the staining pattern appears ring-like.

References:

10. Galloux, M. et al. (2020). Minimal Elements Required for the Formation of Respiratory Syncytial Virus Cytoplasmic Inclusion Bodies *In Vivo* and *In Vitro*. *mBio*. *11*(5): e01202-20.

The authors wish to thank both reviewers again for taking the time to adjudicate our work and greatly value their feedback.